# Impact of Preoperative Food Addiction on Weight Loss and Weight Regain Three Years After Bariatric Surgery

**DOI:** 10.3390/nu17132114

**Published:** 2025-06-26

**Authors:** Fernando Guerrero-Pérez, Natalia Vega Rojas, Isabel Sánchez, Lucero Munguía, Susana Jiménez-Murcia, Cristina Artero, Lucía Sobrino, Claudio Lazzara, Rosa Monseny, Mónica Montserrat, Silvia Rodríguez, Fernando Fernández-Aranda, Nuria Vilarrasa

**Affiliations:** 1Department of Endocrinology and Nutrition, Hospital Universitari de Bellvitge, 08907 Barcelona, Spain; nvegar@bellvitgehospital.cat; 2Biomedical Research Institute of Bellvitge (IDIBELL), L′Hospitalet de Llobregat, 08907 Barcelona, Spain; lsobrino@bellvitgehospital.cat (L.S.); clazzara@bellvitgehospital.cat (C.L.); 3Clinical Psychology Department, Hospital Universitari de Bellvitge, 08907 Barcelona, Spain; isasanchez@bellvitgehospital.cat (I.S.); lmunguia@idibell.cat (L.M.); sjimenez@bellvitgehospital.cat (S.J.-M.); cartero@idibell.cat (C.A.); ffernandez@bellvitgehospital.cat (F.F.-A.); 4Psychoneurobiology of Eating and Addictive Behaviors Group, Neuroscience Program, Bellvitge Biomedical Research Institute (IDIBELL), 08908 Barcelona, Spain; 5CIBER Physiopathology of Obesity and Nutrition (CIBERobn), Health Institute Carlos III, 28029 Madrid, Spain; 6Department of Clinical Sciences, School of Medicine and Health Sciences, University of Barcelona, 08907 Barcelona, Spain; 7General and Gastrointestinal Surgery, Bariatric Surgery Unit, Hospital Universitari de Bellvitge, L′Hospitalet de Llobregat, 08907 Barcelona, Spain; 8Clinical Nutrition Unit, Hospital Universitari de Bellvitge, L′Hospitalet de Llobregat, 08907 Barcelona, Spain; monseny@bellvitgehospital.cat (R.M.); mgil@bellvitgehospital.cat (M.M.); 9Clinical Nurse in Obesity and Bariatric Surgery, Hospital Universitari de Bellvitge, L′Hospitalet de Llobregat, 08907 Barcelona, Spain; srodriguezm@bellvitgehospital.cat; 10CIBER Centro de Investigación Biomédica en Red de Diabetes y Enfermedades Metabólicas Asociadas (CIBERDEM), Health Institute Carlos III, 28029 Madrid, Spain; 11Diabetes and Metabolism Program, Bellvitge Biomedical Research Institute (IDIBELL), 08908 Barcelona, Spain

**Keywords:** food addiction, bariatric surgery, bariatric weight outcomes

## Abstract

*Background*: Food addiction (FA) is prevalent among individuals with severe obesity and has been associated with poorer weight loss (WL) outcomes after dietary interventions. However, its long-term impact after bariatric surgery (BS) remains unclear. *Objective*: This study aimed to evaluate the effect of preoperative FA on WL and weight regain (WR) three years after different BS techniques. *Methods*: An ambispective study was conducted in 165 patients undergoing BS (41.1% sleeve gastrectomy [SG], 13.3% Roux-en-Y gastric bypass [RYGB], and 45.6% hypoabsorptive procedures [HA]). FA was assessed preoperatively using the Yale Food Addiction Scale 2.0. WL outcomes were evaluated at 1 and 3 years postoperatively. Mixed-effects models were used to assess longitudinal changes, adjusted for baseline weight, sex, type 2 diabetes (T2D), and height. *Results*: FA was present in 17.6% of patients. At 3 years, total WL was lower in FA patients compared to those without FA (−27.1% vs. −31.0%; *p* = 0.023), driven by greater WR from nadir (+8.3% vs. +1.7%; *p* = 0.03). The effect was particularly pronounced after RYGB and HA, but not after SG. Nevertheless, a substantial proportion of FA patients (58%) were no longer classified as having obesity at 3 years. The presence of FA was not associated with insufficient WL or lower T2D remission rates. Mixed models confirmed a significant interaction between FA and time, indicating a trend toward reduced WL over time in FA patients. *Conclusions*: Preoperative FA was not associated with a reduced likelihood of achieving satisfactory WL following BS. Our data does not support the use of preoperative FA as a decisive factor in guiding the choice of BS type. Although FA was associated with increased WR over time, clinically meaningful WL was achieved in most patients. Long-term multidisciplinary follow-up remains essential in this subgroup.

## 1. Introduction

Bariatric surgery (BS) remains the most effective and durable intervention for the treatment of severe obesity and its metabolic complications [1]. Nevertheless, insufficient weight loss (IWL) and weight regain (WR) are frequent clinical challenges, affecting up to 40% of patients undergoing restrictive techniques and approximately 20% of those treated with hypoabsorptive (HA) procedures. These figures are clinically relevant, as both IWL and WR may reduce the long-term benefits of BS, including remission of major obesity-related comorbidities. Moreover, they often lead to prolonged pharmacological treatment, revisional procedures, and increased healthcare costs, underscoring the need for early identification and tailored follow-up [1,2,3,4].

Although IWL and WR have traditionally been attributed to patients’ lack of motivation and adherence, it is now recognized that considering weight loss (WL) purely a matter of self-control is an overly simplistic view. Recent cultural and scientific shifts have reframed obesity as a complex chronic disease, influenced by biological, psychological, environmental, and social determinants. Factors such as neuroendocrine regulation, genetic predisposition, metabolic adaptation, emotional dysregulation, socioeconomic status, and exposure to obesogenic environments contribute to this multidimensional understanding, moving beyond the traditional focus on individual willpower [5,6].

The preoperative factors that predispose patients to suboptimal WL outcomes after BS remain incompletely understood and appear to be complex and multifactorial [1]. In addition to poor adherence to dietary guidelines and physical inactivity, potential contributing factors include anatomical surgical failure, metabolic adaptation, and hormonal dysregulation due to foregut exclusion, as well as underlying mental health conditions [1,5]. Additionally, behavioral, psychological, and neurophysiological factors play a critical role in regulating food intake in individuals with obesity [7]. Numerous studies have demonstrated a high prevalence of mental health conditions, such as anxiety and depression, as well as impulsivity and eating disorders among candidates for BS. In this context, food addiction (FA) also emerges as a prevalent and clinically relevant condition in this population [8,9,10].

FA is characterized by a loss of control over the consumption of highly processed, hyperpalatable, or energy-dense foods [11,12]. While FA is not officially recognized as a clinical entity within the Diagnostic and Statistical Manual of Mental Disorders (DSM-5), the topic has been widely discussed in recent years, particularly in relation to WL management and BS [13,14,15]. FA is assessed using the Yale Food Addiction Scale (YFAS), which applies the diagnostic criteria for substance use disorders to eating behaviors [16]. The YFAS identifies individuals exhibiting addiction-like behaviors, such as compulsive intake, intense cravings, and continued consumption despite adverse health consequences [16]. Although FA is not formally included in current diagnostic classifications, the YFAS provides a standardized, symptom-based tool. It is worth noting that the use of an addiction framework to explain eating behavior remains debated, as some authors warn against pathologizing a necessary biological function. Nevertheless, this approach may be useful in subgroups such as BS.

In the general population, FA has been associated with higher caloric intake, greater body weight, and increased fat mass compared to individuals without FA [17]. The presence of FA, as a transdiagnostic construct present in the general population and mental disorders [18], was associated with greater eating symptomatology, general psychopathology, and more dysfunctional personality traits [19], but also with poor therapy response in eating disorder patients living with obesity, namely binge eating disorders [20,21]. Furthermore, FA is particularly prevalent among individuals who are living with obesity and those seeking BS [22,23,24]. A recent meta-analysis of 40 studies, involving 6626 patients, reported a weighted preoperative FA of 32% (95% CI: 27–37%) [25]. FA has also been linked to reduced WL following hypocaloric diets and lifestyle interventions prior to BS [24,25]. Conversely, studies investigating the impact of FA on WL, mainly focused on sleeve gastrectomy (SG) and gastric bypass (RYGB), have yielded inconsistent findings [26,27,28,29,30,31]. While one study found an association between higher post-surgical YFAS scores and WR [27], other studies evaluating short-term outcomes reported no significant differences in percent WL at 6 and 12 months [23,25]. These discrepancies may reflect differences in patient characteristics and follow-up durations across studies. However, it should be noted that WR typically emerges beyond two years postoperatively (particularly following restrictive techniques), and data on mid- and long-term outcomes remains scarce.

In this context, the primary objective of our study was to evaluate whether patients with severe obesity and preoperative FA exhibited lower WL three years after undergoing different BS techniques, including SG, RYGB, duodenal switch (DS), and single anastomosis duodenal switch (SADI-S).

## 2. Materials and Methods

### 2.1. Study Design and Participants

This ambispective, single-center study included consecutive patients with severe obesity admitted for BS at Bellvitge University Hospital between September 2018 and November 2019. The criteria for BS followed the NIH Guidelines (1992) [27], which included the following: (a) age between 18 and 65 years; (b) BMI ≥ 40 kg/m^2^ or >35 kg/m^2^ with major obesity-related comorbidities (arterial hypertension, type 2 diabetes mellitus [T2D], dyslipidemia, obstructive sleep apnea, or severe osteoarthritis); (c) failure to achieve expected weight loss (WL) after hypocaloric diet and/or pharmacologic treatment; (d) willingness and ability to adhere to the BS protocol; and (e) absence of active eating disorders, severe psychiatric conditions, substance abuse, or any unstable medical condition.

### 2.2. Study Variables and Participant Assessment

Demographic, clinical, and biochemical data were obtained from medical records. Definitions for obesity-related comorbidities were as follows: T2D diagnosis was based on glycosylated hemoglobin ≥ 6.5%, two fasting plasma glucose measurements ≥ 126 mg/dL, plasma glucose ≥ 200 mg/dL post-OGTT, random plasma glucose ≥ 200 mg/dL with hyperglycemia symptoms, or current use of glucose-lowering medication [32]. Hypertension was defined as persistent blood pressure ≥ 140/90 mmHg, 24 h ambulatory blood pressure ≥ 130/80 mmHg, or use of antihypertensive medication [33]. Dyslipidemia was diagnosed based on serum lipid abnormalities or use of hypolipidemic agents [34]. Obstructive sleep apnea was diagnosed by polysomnography according to international guidelines [35], and disabling osteoarthritis was defined by documented degenerative joint disease limiting daily activities.

### 2.3. Presurgical Psychological Assessment

All participants underwent two face-to-face semi-structured interviews conducted by experienced clinical psychologists to rule out clinically relevant psychopathology and eating disorders, including binge eating disorder, based on DSM-5 diagnostic criteria. Interviews were conducted by the same team of experienced psychologists using standardized DSM-5 criteria, with diagnostic consensus ensured through regular team meetings and supervision to promote consistency. Additionally, the YFAS 2.0 was administered to assess FA [36]. The validated Spanish version was used [37], which underwent cultural adaptation and psychometric validation for use in Spanish-speaking populations. Minimal clarifications were implemented during administration to ensure cultural appropriateness, under the supervision of the original YFAS author, and following the original scoring guidelines [38] to obtain both a continuous symptom count (0–11) and a binary FA classification (present vs. absent). This approach allowed for the identification of FA without overlapping with formal DSM-5 eating disorder diagnoses.

### 2.4. Pre- and Postsurgical Dietetic Intervention

Before BS, patients followed a standardized dietary protocol, including group education sessions and three individualized counseling visits. Postoperatively, a liquid very low-calorie diet (Optisource^®^) was prescribed for two weeks, followed by a pureed diet for an additional two weeks. During this initial phase, patients received individualized counseling to reinforce instructions, identify barriers (intolerance or satiety), and adjust the plan as needed. Additional support was available on demand through telephone or in-person contact with the dietitian. Texture and variety were progressively reintroduced after the first month. Patients attended dietetic follow-up visits at 3, 6, 12, and 18 months, and were subsequently advised to maintain a hypocaloric Mediterranean diet with annual check-ups. Adherence to dietary recommendations was assessed through regular follow-up visits, during which dietitians reviewed patients’ daily food records as part of routine nutritional monitoring.

### 2.5. Surgical Procedures

Patients underwent one of three surgical techniques: sleeve gastrectomy (SG), Roux-en-Y gastric bypass (RYGB), or duodenal switch/single anastomosis duodeno-ileal bypass with sleeve gastrectomy (DS/SADI-S). SG involved longitudinal resection of the stomach over a 36 French catheter. RYGB included creating a small gastric pouch with a 60 cm biliopancreatic and 150 cm alimentary limb. DS/SADI-S involved SG with duodeno-ileal anastomosis and a 300 cm common channel (SADI-S) or a 200 cm common channel (DS). A multidisciplinary committee determined the appropriate surgical technique based on patient characteristics. Criteria included baseline BMI, the presence of T2D or gastroesophageal reflux disease (GERD), prior abdominal surgery, patient adherence capacity, and pregnancy intention. SG was typically chosen for lower-BMI patients without GERD; RYGB for those with GERD or T2D; and DS/SADI-S for patients with severe obesity or poorly controlled metabolic disease.

### 2.6. Ethical Aspects

Data was collected under standard clinical care. The study complied with the Declaration of Helsinki and Spanish data protection laws. Ethics Committee approval was obtained (PR006/25). As the analysis was retrospective, informed consent was waived.

### 2.7. Statistical Analysis

Categorical variables were expressed as frequencies and percentages, while continuous variables were reported as means ± standard deviations (SDs) or medians and interquartile ranges (IQRs), depending on the distribution. Baseline characteristics between groups were compared using *t*-tests or ANOVA for normally distributed variables and Mann–Whitney U or Kruskal–Wallis tests for non-normally distributed variables. The chi-squared test or Fisher’s exact test was used for categorical variables. To evaluate the effect of time on weight, a linear mixed-effects model was fitted, with time and FA status as fixed effects and patient as a random effect. The interaction model was repeated and adjusted for sex, height, baseline weight, and the presence of T2D. Covariates were selected based on clinical relevance rather than statistical criteria. Other variables were considered but excluded to maintain model parsimony and avoid overfitting. This model accounts for both fixed effects (time, FA status) and random effects (patient-level variability), making it appropriate for modeling individual weight trajectories and handling missing or unbalanced longitudinal data. All statistical analyses were conducted using R version 4.3.3 (R Foundation for Statistical Computing).

## 3. Results

A total of 165 patients were included (75.8% female; mean age 47.6 ± 9.1 years; median BMI 43.5 (39–46) kg/m^2^). Preoperatively, 29 patients (17.6%) met FA criteria. Although baseline body weight was lower in the FA group, no significant differences were observed in other anthropometric parameters, sociodemographic characteristics, comorbidities, or antidepressant use (Table 1).

Regarding BS techniques, 41.1% underwent SG, 13.3% RYGB, and 45.6% HA (DS/SADI-S). SG patients were younger and had a lower proportion of women compared to the RYGB and HA groups. BMI was lower and T2D prevalence higher in the RYGB group. FA prevalence and the distribution of comorbidities other than T2D were similar across surgical techniques (Table 2).

At one-year post-surgery, greater WL was observed after RYGB (−33.32 ± 7.12%) and HA (−33.36 ± 7.51%) compared to SG (−29.06 ± 9.51%; *p* = 0.006). Adjusted analyses showed no significant difference in WL trajectories by FA status (Figure 1). Sixty-two percent of FA patients and 54% of non-FA patients were no longer classified as having obesity (*p* = 0.539). Insufficient WL (<20% total WL) occurred in 17% of FA patients and 11% of non-FA patients (*p* = 0.315). T2D remission rates were similar between groups (90% without FA vs. 71% with FA; *p* = 0.223).

At three-years, greater WL was again observed after HA (−32.10 ± 9.52%) and RYGB (−29.91 ± 9.48%) compared to SG (−27.71 ± 12.4%; *p* = 0.028). Overall WL was lower in FA patients due to greater weight regain (WR) from nadir (+8.3 ± 6.8% vs. +1.7 ± 1.0%; *p* = 0.03). Despite this, 58% of FA patients and 56% of non-FA patients were no longer classified as having obesity (*p* = 0.840). WR was observed in FA patients across all surgical techniques but was particularly pronounced after RYGB (Figure 1).

Mixed models showed a significant interaction between time and FA, indicating progressively less WL over time in patients with FA. Lower body weights were observed in these patients after SG and HA compared to RYGB, suggesting that these techniques may partially mitigate the negative impact of FA. The intraclass correlation coefficient (ICC) of the model was 0.62, indicating substantial variability attributable to between-patient differences (Table 3). No significant associations were observed when using the YFAS 2.0 symptom count as a continuous variable (0–11) instead of the binary FA classification.

## 4. Discussion

To our knowledge, this is the first study to evaluate the impact of presurgical FA on WL and WR over a 3-year period following different BS techniques, including restrictive, mixed, and hypoabsorptive procedures. Our findings indicate that patients with FA exhibited lower overall WL three years after BS, primarily due to greater WR from the weight nadir, particularly following RYGB. However, the magnitude of WR was of limited clinical relevance, and most of the weight variability observed was attributable to inter-individual differences, underscoring the importance of close long-term follow-up in patients undergoing BS.

BS is the most effective intervention for severe obesity and its metabolic complications [1]. However, as observed with other WL approaches, long-term outcomes vary significantly, and WR remains a common issue [1,2,3,4]. WR is a multifactorial phenomenon, with current evidence implicating persistent alterations in adipose tissue immune profiles (obesogenic memory), reductions in resting energy expenditure (metabolic adaptation), impaired lipolysis and lipid oxidation, and gut hormone dysregulation [39]. In addition, physiological factors (hormones such as GLP-1, PYY, and leptin), psychiatric conditions (depression, anxiety), and behavioral aspects (maladaptive eating behaviors, binge eating, and FA) may also contribute to suboptimal WL after BS [1].

Interest in the role of FA in WL outcomes has increased in recent years. FA is highly prevalent among patients seeking BS, with estimates up to one-third of cases [26]. Nevertheless, evidence on its true clinical impact remains limited. A previous study from our group showed that FA predicted poorer WL following a hypocaloric diet and lifestyle intervention prior to BS, suggesting a potential need for specific multidisciplinary approaches in these patients [24].

In the short term, FA does not appear to significantly influence WL after BS. In a study of 195 patients undergoing RYGB, FA was not associated with WL outcomes at 6 or 12 months [28]. Similarly, in a cohort of 166 patients undergoing BS (RYGB or SG), no differences in BMI or percent excess weight loss were observed at 6 and 12 months between patients with and without FA [30]. Consistent with these findings, our 1-year results showed that preoperative FA did not significantly affect WL, with similar WL trajectories across surgical techniques and comparable rates of insufficient WL.

However, it is important to assess WL outcomes beyond the first postoperative year to better capture the risk of WR. In a prospective study of 45 women undergoing SG with a 2-year follow-up, presurgical FA was not significantly associated with WL outcomes [40]. Similarly, our 3-year results confirmed that among SG patients, WL outcomes were comparable regardless of FA status. In contrast, greater WR and slightly lower WL were observed in patients with FA following HA, and especially after RYGB, compared to non-FA patients. Nonetheless, overall WL remained clinically satisfactory across all surgical techniques. According to the 2024 IFSO Consensus on Definitions and Clinical Practice Guidelines, clinically relevant postoperative deterioration is defined as either ≥30% WR from the initial surgical WL or recurrence/worsening of an obesity-related comorbidity that initially indicated surgery [41]. The WR observed in our cohort did not meet these criteria and was therefore not considered clinically relevant.

In our cohort, the greater WR observed among patients undergoing RYGB was initially attributed to the higher prevalence of T2D. However, this difference remained significant even after adjusting for this comorbidity. Upon reviewing individual clinical records, several life events such as job loss, smoking cessation, or marital separation were identified as potential contributors to WR. These observations emphasize the importance of close and ongoing multidisciplinary support after BS.

Despite a tendency toward greater WR, a substantial proportion of FA patients were no longer classified as having obesity. Notably, in SG patients (unlike those who underwent RYGB or HA) no significant differences in WL were observed between FA and non-FA groups at three years. These findings are consistent with those of a recent study reporting poorer WL and greater WR at 5 and 10 years among RYGB patients with FA [42]. Overall, this suggests that while FA may affect long-term weight trajectories, its impact on mid-term WL outcomes may vary depending on the surgical technique and duration of follow-up. A definitive explanation for the comparatively attenuated impact of FA after SG remains uncertain. However, patients selected for SG were generally younger and had fewer obesity-related comorbidities, which may partly account for the more favorable WL. This subgroup also included women with pregnancy intentions, often characterized by high motivation. These factors, along with psychological characteristics and adherence, likely contributed to mitigating the influence of FA in this group.

This study has several strengths, including its prospective design, standardized interventions, use of a validated FA assessment tool, and comprehensive 3-year follow-up. The inclusion of multiple BS techniques and adjustment for key confounders (baseline weight, sex, and T2D) further strengthens the validity of the findings. However, some limitations must be acknowledged. The single-center design may limit the generalizability of our findings. Although the study cohort was recruited between 2018 and 2019, surgical techniques and follow-up protocols at our center have remained largely consistent, supporting the applicability of these findings to current practice. Moreover, the relatively small number of patients with FA (particularly when stratified by surgical techniques) may have reduced statistical power. The diagnosis of FA was based on self-report measures, which may be subject to bias. Additionally, we did not assess several factors known to influence weight loss outcomes after bariatric surgery, such as dietary intake (e.g., food diaries), physical activity levels, hormonal adaptations, or postoperative psychological status. Other potentially relevant variables, such as socioeconomic and educational status, were also not systematically collected, which may limit the identification of additional confounders. Finally, outcomes beyond three years were not evaluated, restricting our ability to assess the long-term impact of FA on weight loss and weight regain.

## 5. Conclusions

Our study demonstrates that preoperative FA does not preclude satisfactory WL following BS. Although FA was associated with a greater tendency toward WR, a significant proportion of FA patients achieved meaningful WL and remission of obesity at 3 years. According to our study, the presence of FA should not determine the choice of surgical technique or favor more aggressive procedures. Instead, surgical decisions should follow each center’s protocol based on patients’ baseline characteristics. We believe that patients with FA do not require a specific surgical protocol. Future research should aim to refine personalized strategies to further optimize results in this specific subgroup.

## Figures and Tables

**Figure 1 nutrients-17-02114-f001:**
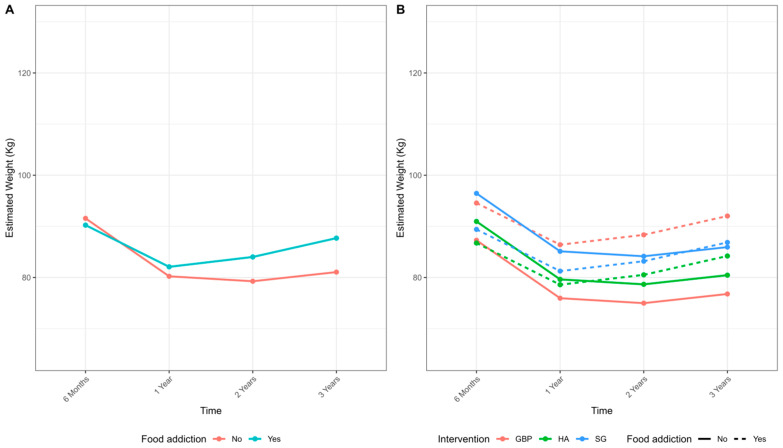
Weight loss (kg) over time by food addiction (FA) status. (**A**) Mean estimated weight at 6 months, 1, 2, and 3 years postoperatively, comparing patients with and without FA. (**B**) Mean estimated weight stratified by surgical procedure: Roux-en-Y gastric bypass (GBP), hypoabsorptive procedures (HA), and sleeve gastrectomy (SG) and FA status (solid line: no FA; dashed line: FA). Sample size at baseline: total n = 165 (FA = 29; non-FA = 136). Mixed-effects model showed a significant interaction between FA and time (*p* = 0.039).

**Table 1 nutrients-17-02114-t001:** Baseline characteristics of patients with and without FA.

	Participants without FA (n = 136)	Participants with FA (n = 29)	*p*
Age (years), means (SD)	51.3 ± 6.5	50.7 ± 7.9	0.145
Female n, (%)	101 (74.3)	22 (75.9)	0.753
Employed n, (%)	66 (48.5)	12 (41.4)	0.400
Physically active n, (%)	58 (42.6)	13 (44.8)	0.879
Weight (kg), means (SD)	119.4 ± 19.4	111.5 ± 20.6	0.042
BMI (kg/m^2^, means (SD)	44.1 ± 5.8	42.1 ± 6.1	0.062
Waist circumference (cm)	130.7 ± 13.9	130.3 ± 17.7	0.732
Hypertension n, (%)	58 (42.6)	16 (55.2)	0.244
Dyslipidemia n, (%)	48 (35.3)	11 (37.9)	0.830
Obstructive sleep apnea n, (%)	62 (45.6)	14 (48.3)	0.844
T2D n, (%)	31 (22.8)	10 (34.5)	0.713
Glycosylated hemoglobin (%), means (SD)	5.9 ± 1.1	5.8 ± 0.7	0.844
GLP-1 n, (%)	11 (8.1)	3 (10.3)	0.167
SGLT2 n, (%)	7 (5.1)	1 (3.4)	0.782
Insulin n, (%)	9 (6.6)	1 (3.4)	0.786
Antidepressant drugs n, (%)	35 (25.7)	10 (34.5)	0.374

BMI, body mass index; FA, food addiction; GLP-1, glucagon-like peptide-1 agonists receptor; SGLT2, sodium–glucose cotransporter-2 inhibitors; T2D, type 2 diabetes.

**Table 2 nutrients-17-02114-t002:** Baseline characteristics of patients according to surgical procedures.

	SG (n = 68)	GBP (n = 22)	HA (n = 75)	*p*
FA (%)	21.2	27.2	12	0.177
Age (years), means (SD)	43.5 ± 5.2	52.6 ± 1.8	51.13 ± 0.9	0.005
Female n, (%)	44 (64.7)	20 (90.9)	61 (81.3)	0.014
Employed n, (%)	27 (37.9)	12 (54.5)	39 (52)	0.279
Works out n, (%)	24 (35.3)	13 (59.1)	35 (46.7)	0.114
Weight (kg), means (SD)	119.5 ± 2.7	106.5 ± 3.9	120.1 ± 1.9	0.013
BMI (kg/m^2^), means (SD)	43.6 ± 0.8	39.8 ± 0.9	44.9 ± 0.6	0.001
Waist circumference (cm), means (SD)	131.7 ± 16.9	123.9 ± 11.4	131.1 ± 12.6	0.080
Hypertension n, (%)	24 (35.3)	11 (50)	39 (52)	0.117
Dyslipidemia n, (%)	20 (29.4)	6 (27.3)	33 (44)	0.129
Obstructive sleep apnea n, (%)	31 (45.6)	8 (36.4)	38 (50.7)	0.484
T2D (%)	16.1	40.9	28.0	0.046
Glycosylated hemoglobin (%), means (SD)	5.7 ± 0.9	5.9 ± 0.9	6.0 ± 1.1	0.190
GLP-1 (%)	8.8	13.6	6.6	0.582
SGLT2 (%)	5.8	13.6	1.3	0.054
Insulin (%)	4.4	13.6	5.3	0.271
Antidepressant drugs, n, (%)	19 (27.9)	6 (27.3)	20 (26.7)	0.974

BMI, body mass index; FA, food addiction; GLP-1, glucagon-like peptide-1 agonists receptor; SGLT2, sodium–glucose cotransporter-2 inhibitors; T2D, type 2 diabetes.

**Table 3 nutrients-17-02114-t003:** Mixed models evolution of Weight.

Predictors	Estimates	SE	CI	*p*	Estimates	Std. Error	CI	*p*	Estimates	SE	CI	*p*
Intercept	16.37	4.51	7.51–25.23	<0.001	15.32	4.40	6.67–23.96	0.001	−14.05	17.26	−47.96–19.85	0.416
1 Year vs. 6 months	−10.79	0.69	−12.15–−9.43	<0.001	−11.33	0.75	−12.81–−9.85	<0.001	−11.33	0.75	−12.81–−9.85	<0.001
2 Years vs. 6 months	−11.32	0.70	−12.69–−9.94	<0.001	−12.29	0.76	−13.79–-10.80	<0.001	−12.30	0.76	−13.79–−10.81	<0.001
3 Years vs. 6 months	−9.26	0.72	−10.68–−7.84	<0.001	−10.50	0.79	−12.05–−8.95	<0.001	−10.51	0.79	−12.06–−8.96	<0.001
Baseline Weight	0.63	0.04	0.56–0.71	<0.001	0.61	0.04	0.54–0.68	<0.001	0.52	0.05	0.43–0.62	<0.001
FA [Yes vs. No]	1.81	1.87	−1.87–5.49	0.334	5.50	3.97	−2.29–13.29	0.166	6.80	3.91	−0.87–14.48	0.082
Intervention [HA vs. RYGB]					1.97	2.53	−3.01–6.94	0.438	3.72	2.55	−1.28–8.72	0.145
Intervention [SG vs. RYGB]					8.33	2.59	3.26–13.41	0.001	9.16	2.60	4.05–14.27	<0.001
Time [1 year] × FA [Yes]					2.97	1.78	−0.53–6.47	0.096	3.02	1.78	−0.48–6.52	0.091
Time [2 years] × FA [Yes]					5.48	1.79	1.96–8.99	0.002	5.48	1.79	1.96–8.99	0.002
Time [3 years] × FA [Yes]					6.82	1.84	3.22–10.43	<0.001	6.80	1.84	3.20–10.41	<0.001
Intervention [HA] × FA [Yes]					−7.38	5.01	−17.22–2.47	0.142	−10.58	5.02	−20.43–−0.73	0.035
Intervention [SG] × FA [Yes]					−12.11	4.69	−21.33–−2.89	0.010	−13.41	4.61	−22.46–−4.35	0.004
Male vs. Female									1.76	2.21	−2.58–6.10	0.425
Height									0.23	0.12	−0.00–0.46	0.051
T2D [Yes vs. No]									1.46	1.62	−1.72–4.63	0.368

CI, confidence interval; FA, food addiction; HA, hypoabsorptive technique (refers to duodenal switch or single anastomosis duodeno-ileal bypass), RYGB, Roux-en-Y gastric bypass; SE, standard error; SG, sleeve gastrectomy; T2D, type 2 diabetes.

## Data Availability

The data supporting the findings of this study is available upon reasonable request from the corresponding author. The data is not publicly available due to privacy and ethical restrictions.

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
