# Peer review of "Impact of Preoperative Food Addiction on Weight Loss and Weight Regain Three Years After Bariatric Surgery"

_nutrients, 2025, doi:10.3390/nu17132114_

Round 1
Reviewer 1 Report
Comments and Suggestions for Authors
- The authors mention that "insufficient weight loss (IWL) and weight regain (WR) are frequent clinical challenges, affecting up to 40% of patients undergoing restrictive techniques and approximately 20% of those treated with hypoabsorptive (HA) procedures." Could the author expound upon the clinical relevance of these percentages? What are the consequences for patient health and healthcare resources when IWL or WR occur at these frequencies?
- The authors argue that viewing weight reduction solely as a matter of self-control is an overly simplistic perspective. Could they elaborate on the cultural and medical transformations that have contributed to a more comprehensive understanding of obesity and weight control while discussing the various factors that contribute to this understanding?
- Food Addiction (FA) is a crucial notion in this research. The authors observe that it is not formally acknowledged in the DSM-5. What effect may the absence of formal recognition have on the diagnostic consistency and therapeutic use of the YFAS, particularly across diverse healthcare environments or cultural contexts?
- The YFAS employs diagnostic criteria for drug use disorders. Could the authors succinctly address the issues or disputes over the application of an addiction paradigm to food intake and how the study's emphasis on FA in bariatric surgery patients may enhance this discourse?
- The authors reference a meta-analysis indicating a weighted preoperative prevalence of FA at 32%. How does this prevalence relate to other mental health disorders or behavioral variables often assessed in candidates for bariatric surgery? Does it indicate that FA is a notably significant comorbidity?
- The authors acknowledge the contradictory results in prior research evaluating the influence of FA on weight reduction after SG and RYGB, particularly regarding short-term outcomes. What particular constraints or methodological variances (e.g., patient demographics, FA evaluation, follow-up period) in those studies may elucidate these discrepancies?
- The primary purpose of the present research is to assess reduced weight loss three years after surgery. What makes the three-year milestone particularly significant for evaluating the influence of FA on WL outcomes after BS, especially considering the authors' note that WR "usually arises beyond two years postoperatively"?
- This is a proposed, single-center investigation. This methodology facilitates controlled data collection; nevertheless, what are the possible constraints of single-center research regarding the generalizability of the results to a broader bariatric surgery population?
- The research included patients hospitalized from September 2018 to November 2019. Given the rapid advancement in bariatric surgery methodologies and perioperative management, do the authors foresee any alterations in practice since that period that might affect the applicability of these results to contemporary patients?
- One inclusion criterion is "absence of active eating disorders, severe psychiatric conditions, substance abuse, or any unstable medical condition." Considering that FA is evaluated using YFAS, how was the differentiation established between a "food addiction" diagnosis via YFAS and other "active eating disorders" (e.g., Binge Eating Disorder) that were designated as exclusion criteria? This seems essential for elucidating your participant cohort.
- The authors gathered demographic, clinical, and biochemical data. Were other patient factors (e.g., socioeconomic status, educational attainment, particular drug use) contemplated for collection but ultimately excluded, and if so, what was the justification?
- Patients participated in two in-person semi-structured interviews to exclude psychopathology and eating disorders. What measures were used to guarantee inter-rater reliability among the clinical psychologists doing these interviews?
- The authors used YFAS 2.0 and its certified Spanish counterpart. Were any cultural changes or considerations implemented in the application of the YFAS for the Spanish-speaking community beyond merely providing a translated version to guarantee its suitability?
- Patients adhered to a "standardized dietary protocol" and participated in "group education sessions" as well as "individualized counseling appointments." What methods were used to monitor adherence to this regimen before and after surgery? Were any objective assessments of dietary adherence obtained?
- Postoperatively, a "very low-calorie liquid diet" was suggested for two weeks, followed by pureed food. What particular advice or assistance was offered to patients during this first hard dietary period to guarantee adherence and address any difficulties?
- A multidisciplinary committee determined the suitable surgical procedure. What were the main criteria or algorithms used by this committee to determine the suitability of SG, RYGB, and DS/SADIS for specific patients? This information is crucial for comprehending the allocation of patients to various procedures.
- The authors used a linear mixed-effects model, including the patient as a random variable and time and FA as fixed factors. What prompted the use of a linear mixed-effects model over other longitudinal research techniques, and what distinct benefits does it provide for examining weight loss over time while accommodating individual patient variability?
- The model was modified to account for sex, height, baseline weight, and the presence of type 2 diabetes (T2D). Did the authors contemplate correcting for any other confounding factors but ultimately choose not to? What was the justification for their exclusion?
The English might be enhanced to convey the study more effectively.
Author Response
We would like to thank the reviewers and the editorial team for their thorough evaluation of our manuscript and for their valuable comments. On behalf of all co-authors, we express our appreciation for the time and effort dedicated to reviewing our work. The issues raised are of great importance and have helped us improve the clarity and quality of the manuscript. We have thoroughly revised the entire document, including the English language, as well as all tables and the figure. Below, we address each comment point by point.
REVIEWER 1:
Comments and Suggestions for Authors
- The authors mention that "insufficient weight loss (IWL) and weight regain (WR) are frequent clinical challenges, affecting up to 40% of patients undergoing restrictive techniques and approximately 20% of those treated with hypoabsorptive (HA) procedures." Could the author expound upon the clinical relevance of these percentages? What are the consequences for patient health and healthcare resources when IWL or WR occur at these frequencies?
Response: We thank the reviewer for this observation. We agree that it is essential to contextualize the clinical relevance of the reported rates of insufficient weight loss (IWL) and weight regain (WR) following bariatric surgery. These phenomena are associated with reduced long-term effectiveness of the procedure in terms of metabolic and cardiovascular outcomes. Specifically, they may lead to suboptimal remission or persistence of major obesity-related comorbidities, including type 2 diabetes mellitus, arterial hypertension, dyslipidemia, and obstructive sleep apnea. Furthermore, patients experiencing IWL or WR often require extended pharmacological therapy, additional psychological or nutritional interventions, and in some cases revisional surgery. These factors not only impact patient health and quality of life but also result in greater healthcare resource utilization and economic burden. We have expanded on this aspect in the Introduction section (lines 53–57).
- The authors argue that viewing weight reduction solely as a matter of self-control is an overly simplistic perspective. Could they elaborate on the cultural and medical transformations that have contributed to a more comprehensive understanding of obesity and weight control while discussing the various factors that contribute to this understanding?
Response: We appreciate the reviewer’s comment. In recent decades, there has been a significant shift in both medical and cultural perspectives on obesity. Traditionally, weight control was primarily attributed to individual responsibility and self-discipline. However, advances in neurobiology, endocrinology, and behavioral sciences have revealed that obesity is a complex, multifactorial chronic disease. This broader understanding recognizes the interplay between genetic predisposition, hormonal regulation, metabolic adaptation, environmental factors, psychological traits (e.g., impulsivity, emotional regulation), and sociocultural influences (e.g., food marketing, urban design, socioeconomic status). At a policy level, there is increasing advocacy for reframing obesity as a public health issue rather than a personal failure, as reflected in recent international consensus statements. We have elaborated on this conceptual evolution in the Introduction section (lines 60-65).
- Food Addiction (FA) is a crucial notion in this research. The authors observe that it is not formally acknowledged in the DSM-5. What effect may the absence of formal recognition have on the diagnostic consistency and therapeutic use of the YFAS, particularly across diverse healthcare environments or cultural contexts?
Response: Indeed, the absence of formal recognition of Food Addiction (FA) in the DSM-5 presents both conceptual and practical challenges. Although FA is not an official diagnosis, the Yale Food Addiction Scale (YFAS) offers a standardized tool grounded in DSM-based criteria for substance use disorders and has demonstrated good psychometric properties across diverse populations and cultural contexts.
However, the lack of official diagnostic status may affect the consistency of its clinical interpretation and limit its therapeutic applicability in settings where FA is not widely recognized. Despite these limitations, there is growing evidence supporting FA as a distinct construct, with highly processed foods, particularly those rich in added sugars and fats, most strongly implicated in addictive-like eating behaviors (Gordon et al., 2018; Gearhardt et al., 2023). The increasing use of the FA construct in international research highlights its relevance and supports the need for continued investigation, even in the absence of formal nosological classification. We have reflected on these considerations (Introduction, lines 85-86).
- The YFAS employs diagnostic criteria for drug use disorders. Could the authors succinctly address the issues or disputes over the application of an addiction paradigm to food intake and how the study's emphasis on FA in bariatric surgery patients may enhance this discourse?
Response: We thank the reviewer for this observation. The application of an addiction paradigm to food intake remains a subject of debate. Critics argue that likening eating to substance use may pathologize a fundamental behavior and lacks a clearly defined addictive substance. However, growing neurobiological and behavioral evidence supports the notion that certain highly palatable foods, particularly those rich in sugar and fat, can activate reward pathways similar to those involved in substance use disorders (SUDs).
The Yale Food Addiction Scale (YFAS) applies SUD criteria to eating behaviors, offering a dimensional approach to identify addiction-like patterns such as loss of control, craving, and persistent use despite negative consequences. By focusing on FA in bariatric surgery candidates, a population in which these behaviors are relatively prevalent, our study provides empirical data that may help clarify the relevance and clinical utility of this construct. We have incorporated this perspective in the Introduction (lines 86-89).
- The authors reference a meta-analysis indicating a weighted preoperative prevalence of FA at 32%. How does this prevalence relate to other mental health disorders or behavioral variables often assessed in candidates for bariatric surgery? Does it indicate that FA is a notably significant comorbidity?
Response: We thank the reviewer for this observation. Indeed, the preoperative prevalence of food addiction (FA) is comparable to that of other mental health conditions frequently observed in bariatric surgery candidates, such as anxiety, depression, or binge eating disorder. This highlights the clinical significance of FA as a relevant behavioral comorbidity in this population. We have briefly addressed this point in the Introduction (lines 71-76).
6-The authors acknowledge the contradictory results in prior research evaluating the influence of FA on weight reduction after SG and RYGB, particularly regarding short-term outcomes. What particular constraints or methodological variances (e.g., patient demographics, FA evaluation, follow-up period) in those studies may elucidate these discrepancies?
Response: The inconsistent findings across previous studies on the impact of food addiction (FA) on outcomes after sleeve gastrectomy (SG) and Roux-en-Y gastric bypass (RYGB) may be explained by several methodological differences. Notably, many of these studies relied on short follow-up periods, typically limited to 6 or 12 months, during which weight loss tends to be at its maximum regardless of behavioral or psychological factors. This may obscure the potential impact of FA, which might become more evident during the weight stabilization or regain phase. Additional variability arises from differences in sample size, baseline patient characteristics (e.g., age, BMI, psychiatric comorbidities) and lack of adjustment for relevant confounders. We have briefly acknowledged these limitations in the Introduction (lines 104-105).
7-The primary purpose of the present research is to assess reduced weight loss three years after surgery. What makes the three-year milestone particularly significant for evaluating the influence of FA on WL outcomes after BS, especially considering the authors' note that WR "usually arises beyond two years postoperatively"?
Response: The three-year follow-up point was chosen because it captures a clinically relevant time frame in the postoperative course of bariatric surgery. While the majority of weight loss occurs within the first 6–12 months, weight stabilization typically follows during the second year, and weight regain (WR) often begins to emerge thereafter, particularly in patients with behavioral vulnerabilities. Thus, assessing outcomes at three years allows us to evaluate not only the initial weight loss but also the onset and trajectory of WR. This period is especially pertinent for understanding the long-term impact of food addiction, which may not significantly affect short-term outcomes but could influence weight trajectories once the intensive early follow-up period ends.
8-This is a proposed, single-center investigation. This methodology facilitates controlled data collection; nevertheless, what are the possible constraints of single-center research regarding the generalizability of the results to a broader bariatric surgery population?
Response: Thank you for your comment. We agree that single-center studies may limit the generalizability of findings due to potential center-specific characteristics, such as patient selection criteria, surgical expertise, follow-up protocols, and population demographics. However, this design also offers strengths, including uniform assessment procedures, consistent surgical techniques, and standardized perioperative care, all of which enhance internal validity and reduce variability. We have acknowledged this limitation in the Discussion section (lines 308-309).
9-The research included patients hospitalized from September 2018 to November 2019. Given the rapid advancement in bariatric surgery methodologies and perioperative management, do the authors foresee any alterations in practice since that period that might affect the applicability of these results to contemporary patients?
Response: Although bariatric surgery has continued to evolve in recent years, particularly with increased adoption of enhanced recovery protocols and technological refinements, the fundamental principles of the surgical techniques analyzed in our study—SG, RYGB, and HA procedures—have remained consistent in terms of anatomical modification and expected metabolic impact. Likewise, the criteria for patient selection, preoperative evaluation (including psychological assessment), and postoperative nutritional follow-up at our center have not changed substantially since the study period. Therefore, we believe that our findings remain applicable to current clinical practice. This point has been briefly noted in the Discussion (lines 309-313).
10-One inclusion criterion is "absence of active eating disorders, severe psychiatric conditions, substance abuse, or any unstable medical condition." Considering that FA is evaluated using YFAS, how was the differentiation established between a "food addiction" diagnosis via YFAS and other "active eating disorders" (e.g., Binge Eating Disorder) that were designated as exclusion criteria? This seems essential for elucidating your participant cohort.
Response: To ensure diagnostic clarity, all patients underwent two face-to-face semi-structured interviews conducted by experienced clinical psychologists from the Department of Clinical Psychology and the Eating Disorders Unit of our university hospital. These evaluations were based on DSM-5 criteria and were used to exclude participants with any current EDs, such as binge eating disorder, bulimia nervosa, or night eating syndrome, as well as other severe psychiatric conditions.
In contrast, FA was assessed dimensionally using YFAS 2.0, which evaluates addiction-like eating behaviors but does not constitute a formal clinical diagnosis under current nosological systems. Therefore, patients could meet criteria for FA on the YFAS without fulfilling diagnostic thresholds for an active ED. We have clarified this distinction in the Methods section (lines 147-148).
- The authors gathered demographic, clinical, and biochemical data. Were other patient factors (e.g., socioeconomic status, educational attainment, particular drug use) contemplated for collection but ultimately excluded, and if so, what was the justification?
Response: Thank you for this point. While our study collected comprehensive demographic, clinical, and biochemical data, additional variables such as educational attainment, detailed socioeconomic status, and non-psychotropic drug use were not systematically recorded. This decision was primarily due to the observational nature of the study and the reliance on routinely available clinical data during the standard preoperative evaluation. Although these variables may be relevant in influencing long-term weight outcomes, their exclusion represents a limitation, which we now acknowledge in the Discussion (lines 317-319).
12-Patients participated in two in-person semi-structured interviews to exclude psychopathology and eating disorders. What measures were used to guarantee inter-rater reliability among the clinical psychologists doing these interviews?
Response: We appreciate this suggestion. All psychological assessments were conducted by the same team of senior clinical psychologists specialized in eating and addictive behaviors, with extensive experience in bariatric evaluations. Although no formal inter-rater reliability coefficients were calculated, diagnostic concordance was ensured through standardized use of DSM-5 criteria, regular case discussion meetings, and supervision by a lead psychologist to harmonize clinical criteria and interpretation. We have clarified this aspect in the Methods section (lines 138-141).
13-The authors used YFAS 2.0 and its certified Spanish counterpart. Were any cultural changes or considerations implemented in the application of the YFAS for the Spanish-speaking community beyond merely providing a translated version to guarantee its suitability?
Response: The Spanish version of the Yale Food Addiction Scale 2.0 (YFAS 2.0) used in our study is not a direct translation but a culturally adapted and psychometrically validated instrument for Spanish-speaking populations, as described by Granero et al. (2018). The adaptation process included forward and backward translation, cultural adjustments to enhance comprehension of item content and food-related examples, and formal validation of its psychometric properties.
Importantly, this version was developed under the guidance of Dr. Ashley Gearhardt, the original author of the YFAS, ensuring fidelity to the original conceptual structure. Minimal clarifications were made during administration to account for local cultural nuances, without altering the core content or scoring. This has been clarified in the Methods section (lines 142-145).
14-Patients adhered to a "standardized dietary protocol" and participated in "group education sessions" as well as "individualized counseling appointments." What methods were used to monitor adherence to this regimen before and after surgery? Were any objective assessments of dietary adherence obtained?
Response: Thank you for your comment. Patients followed a standardized dietary protocol before and after surgery, which included group education sessions and individualized counseling by registered dietitians. Adherence was monitored through scheduled follow-up visits (at 3, 6, 12, and 18 months), during which dietitians systematically reviewed patients’ daily food records. These dietary logs served as an objective tool to assess compliance with nutritional recommendations, in conjunction with weight trends and behavioral observations. We have clarified this aspect in the Methods section (lines 159-161).
15-Postoperatively, a "very low-calorie liquid diet" was suggested for two weeks, followed by pureed food. What particular advice or assistance was offered to patients during this first hard dietary period to guarantee adherence and address any difficulties?
Response: During the immediate postoperative period, patients received structured support from the dietetic team to ensure adherence to the very low-calorie liquid diet and the subsequent pureed phase. This included individualized counseling sessions to reinforce dietary instructions, identify potential barriers (e.g., intolerance, nausea, early satiety), and provide strategies to manage them. In addition, patients had access to telephone or in-person support on demand during this phase to address emergent concerns. This has been clarified in the Methods section (lines 153-156).
16-A multidisciplinary committee determined the suitable surgical procedure. What were the main criteria or algorithms used by this committee to determine the suitability of SG, RYGB, and DS/SADIS for specific patients? This information is crucial for comprehending the allocation of patients to various procedures.
Response: The decision regarding the type of bariatric surgery was made by a multidisciplinary committee based on clinical guidelines and individualized patient characteristics. Key criteria included baseline BMI, presence and severity of type 2 diabetes, gastroesophageal reflux disease (GERD), previous abdominal surgery, patient adherence capacity, and in selected cases, reproductive plans. In general, SG was preferred in patients with lower BMI and no GERD, RYGB in those with GERD or T2D requiring a malabsorptive effect, and DS/SADIS in patients with higher BMI and/or severe metabolic disease. We have clarified these criteria in the Methods section (lines 170-174).
17-The authors used a linear mixed-effects model, including the patient as a random variable and time and FA as fixed factors. What prompted the use of a linear mixed-effects model over other longitudinal research techniques, and what distinct benefits does it provide for examining weight loss over time while accommodating individual patient variability?
Response: A linear mixed-effects model was chosen because it is well suited for analyzing longitudinal data with repeated measures, especially in real-world clinical contexts where follow-up data may be incomplete or unequally spaced. This approach allows modeling both fixed effects (time, FA status) and random effects (individual patient variability), thus capturing individual weight trajectories while estimating overall group-level effects. Compared to repeated measures ANOVA, linear mixed models are more robust in the presence of missing data and do not require assumptions such as sphericity. Moreover, they offer greater flexibility than Generalized Estimating Equations or Covariance Pattern Models by allowing for random intercepts and slopes, which accounts for varying baselines and rates of change across patients. This rationale has been added to the Methods section (lines 190-193).
18-The model was modified to account for sex, height, baseline weight, and the presence of type 2 diabetes (T2D). Did the authors contemplate correcting for any other confounding factors but ultimately choose not to? What was the justification for their exclusion?
Response: The variables included in the adjusted linear mixed-effects model (sex, height, baseline weight, and the presence of type 2 diabetes) were selected based on their known clinical relevance in predicting weight loss outcomes after bariatric surgery. Their inclusion was determined a priori by expert consensus, not through statistical selection methods. Other potential confounders (age, pharmacological treatment, or surgical history) were considered but excluded to preserve model parsimony and prevent overfitting, especially given the sample size. We have clarified this point in the Statistical Analysis section (lines 188-190).
Reviewer 2 Report
Comments and Suggestions for Authors
This well-designed prospective study provides valuable insights into the impact of preoperative food addiction (FA) on long-term bariatric surgery outcomes. The findings are clinically relevant, particularly the observation that most FA patients achieved meaningful weight loss (WL) despite higher weight regain (WR). However, several points require clarification. The term “limited clinical relevance” for WR should be defined with supporting references. The small FA subgroup should be acknowledged as a limitation, especially in subgroup analyses. The use of a binary YFAS classification may mask symptom variability—consider discussing the potential role of continuous scores. Additionally, causal language in the abstract should be revised to reflect the observational nature of the study. Minor issues include typographical corrections, clearer figure legends (especially for statistical significance and sample size), consistent abbreviation use, and more discussion on why sleeve gastrectomy outcomes appear less influenced by FA. With these minor revisions, the manuscript will be suitable for publication.
Author Response
We would like to thank the reviewers and the editorial team for their thorough evaluation of our manuscript and for their valuable comments. On behalf of all co-authors, we express our appreciation for the time and effort dedicated to reviewing our work. The issues raised are of great importance and have helped us improve the clarity and quality of the manuscript. We have thoroughly revised the entire document, including the English language, as well as all tables and the figure. Below, we address each comment point by point.
REVIEWER 2:
Comments and Suggestions for Authors
1-The term “limited clinical relevance” for WR should be defined with supporting references.
Response: We thank the reviewer for this comment. As defined by the International Federation for the Surgery of Obesity and Metabolic Disorders (IFSO) Consensus on Definitions and Clinical Practice Guidelines 2024 (Salminen, P., Kow, L., Aminian, A. et al. IFSO Consensus on Definitions and Clinical Practice Guidelines for Obesity Management—an International Delphi Study. OBES SURG 34, 30–42 (2024). https://doi.org/10.1007/s11695-023-06913-8), clinically relevant weight regain (WR) after bariatric surgery is defined as either a ≥30% regain of the initial weight lost or the recurrence/worsening of an obesity-related comorbidity that was an indication for surgery. In our study, the observed WR did not meet these criteria and was therefore considered not clinically relevant. This has now been clarified in the manuscript (lines 280-284, reference 41).
2-The small FA subgroup should be acknowledged as a limitation, especially in subgroup analyses.
Response: We agree with the reviewer’s observation. The relatively small size of the FA subgroup may limit the statistical power of subgroup analyses and the generalizability of the findings. This has now been explicitly acknowledged as a limitation in the revised manuscript (lines 311-313).
3-The use of a binary YFAS classification may mask symptom variability—consider discussing the potential role of continuous scores.
Response: While the binary classification provided by the Yale Food Addiction Scale (YFAS 2.0) is useful for identifying clinically relevant cases, we agree that it may not fully capture the spectrum of symptom severity. To address this, we also explored the YFAS symptoms count as a continuous variable in our cohort. However, no significant associations were found between symptom severity and weight loss outcomes. These findings suggest that, in this sample, the binary approach adequately reflected relevant clinical differences. This has now been noted in the Results section (lines 236–237).
4-Additionally, causal language in the abstract should be revised to reflect the observational nature of the study.
Response: We thank the reviewer for this observation. We agree that causal language should be avoided given the ambispective and observational design of our study. Accordingly, we have revised the abstract to ensure that the wording reflects associations rather than causality (lines 29, 41-44).
5-Minor issues include typographical corrections, clearer figure legends (especially for statistical significance and sample size).
Response: We have carefully revised the manuscript for typographical issues and have updated the figure legends to improve clarity. Specifically, we now provide details regarding statistical significance and sample size in the revised legend of Figure 1 (lines 226-230).
6-More discussion on why sleeve gastrectomy outcomes appear less influenced by FA. With these minor revisions, the manuscript will be suitable for publication.
Response: While a definitive explanation for the relatively favorable outcomes among patients undergoing sleeve gastrectomy (SG) remains uncertain, several factors may contribute. Patients selected for SG in our cohort were generally younger and had fewer obesity-related comorbidities, as per protocol. This subgroup also included women planning pregnancy, who typically exhibit high motivation and adherence to lifestyle changes. In contrast, high-risk patients who underwent SG as a staged first-step procedure before conversion to a hypoabsorptive technique were excluded. These factors may have attenuated the impact of food addiction (FA) in the SG group. Additionally, long-term weight loss is shaped by multiple elements beyond surgical technique, including individual psychological traits and motivation. We have incorporated this interpretation in the Discussion section (lines 298-303).